# The Nutritional and Functional Properties of Protein Isolates from Defatted Chia Flour Using Different Extraction pH

**DOI:** 10.3390/foods12163046

**Published:** 2023-08-14

**Authors:** Etty Syarmila Ibrahim Khushairay, Ma’aruf Abd Ghani, Abdul Salam Babji, Salma Mohamad Yusop

**Affiliations:** 1Department of Food Sciences, Faculty of Science and Technology, Universiti Kebangsaan Malaysia, Bangi 43600, Selangor, Malaysia; p103278@siswa.ukm.edu.my (E.S.I.K.); daging@ukm.edu.my (A.S.B.); 2Faculty of Fisheries and Food Science, Universiti Malaysia Terengganu, Kuala Terengganu 21030, Terengganu, Malaysia; maaruf@umt.edu.my; 3Innovative Centre for Confectionery Technology (MANIS), Faculty of Science and Technology, Universiti Kebangsaan Malaysia, Bangi 43600, Selangor, Malaysia

**Keywords:** alkaline extraction pH, defatted chia flour, functional properties, nutritional properties, protein isolates

## Abstract

This study aims to determine the effects of various alkaline pHs on the nutritional and functional properties of protein isolated from defatted chia flour (DCF). The DCF isolated using alkali extraction method at pH 8.5, 10.0, and 12.0 were coded as CPI-8.5, CPI-10.0, and CPI-12.0, respectively. The highest extraction yield and protein recovery yield was demonstrated by CPI-12.0 (19.10 and 59.63%, respectively), with a total protein content of 74.53%, and glutelin showed the highest portion (79.95%). The CPI-12.0 also demonstrated the most elevated essential (36.87%), hydrophobic (33.81%), and aromatic (15.54%) amino acid content among other samples. The DCF exhibited the highest water (23.90 gg^−1^) and oil (8.23 gg^−1^) absorption capacity, whereas the CPI-8.5 showed the highest protein solubility (72.31%) at pH 11. DCF demonstrated the highest emulsifying capacity at pH 11 (82.13%), but the highest stability was shown at pH 5 (82.05%). Furthermore, CPI-12.0 at pH 11 shows the highest foaming capacity (83.16%) and stability (83.10%). Despite that, the CPI-10.0 manifested the highest antioxidant capacity (DPPH: 42.48%; ABTS: 66.23%; FRAP: 0.19), as well as ACE-I (35.67%). Overall, the extraction pH had significant effects in producing chia protein isolates (CPI) with improved nutritional and functional qualities.

## 1. Introduction

The role of protein in a healthy diet is becoming more generally acknowledged as a consequence of growing food demand induced by population expansion, as well as a need for sustainable food alternatives generated by socioeconomic changes. The search for natural protein sources, including bioactive substances, to create functional foods and nutraceuticals with high nutritional effects and concurrent health benefits is intensifying [1]. Dietary protein is a source of exogenous peptides that can perform similar regulatory functions in our bodies as endogenous peptides to govern our endocrine and neurological systems [2]. Aside from the growth of vegan, vegetarian, and flexitarian populations, the allergenicity of animal proteins and the versatility of plant proteins in a wide range of natural product manufacturing have propelled the usage of plant proteins as a part of healthy diet [3] as well as functional ingredients in food products [4]. Furthermore, the isolated plant active compounds have drawn more interest because they are thought to be less hazardous and have lower toxicity [5]. In recent years, public interest in using plant protein as a better option than animal protein-based products have grown due to health, ethical, and religious considerations [6]. Several plants, including neglected and underused crops with high protein content, have been researched as possible protein derivations in order to increase the available list of protein sources.

Chia seeds (*Salvia hispanica* L.) are an ancient pseudo-cereal that has evolved into the golden and super seeds of the twenty-first century due to their superior nutritional content and techno-functional qualities [7]. It has acquired popularity as a great source of polyunsaturated fatty acids (namely 3-alpha-linolenic acid and 6-alpha-linoleic acid), which have been demonstrated to improve the cardiovascular system [8]. The enormous demand for chia oil in the marketplace, both for consumption and industrial purposes, has propelled the chia oil industry’s expansion. A considerable portion of the underutilized non-fatty fraction known as defatted chia flour (DCF) is discarded along with the large-scale manufacture of chia oil. Besides fats, chia seed is rich in protein (19–27%), much more significant than other typical food crops, including wheat (~14%), corn (~14%), rice (~14%), oats (~15.3%), and amaranth (~14.8%) [9]. The process of defatting and mucilage removal let the protein increase by up to 35.5% [10]. According to Khursheed et al. [11] and Ullah et al. [12], chia seed has rich amino acids profiles with good balance of essential and non-essential amino acids, proving its excellent protein quality, comparatively to various other grains [13].

The versatility or utilization of protein isolates functional qualities in food composition is mainly determined by the variety of functional capabilities, which are highly influenced by the structural conformation of their proteins [14]. The selection of extraction methods and conditions is crucial in food processing because they can significantly affect the end products’ structure and nutritional and functional characteristics [15]. According to Lopez et al. [16], pH is one of the most critical factors influencing protein conformation because it changes the charges and degree of the unfolding of the proteins. Isoelectric precipitation is a typical method for isolating alkali-extracted plant proteins and preparing high-purity protein-rich products [17]. Alkali and acid treatments during protein extraction through isoelectric precipitation, particularly, are known to cause structural alterations in proteins [18]. The goal of this study was to determine the effects of alkaline extraction pH used on the nutritional and functional characteristics of the chia protein isolates. The finding obtained from this study could potentially broaden the knowledge, enhance the exploitation of DCF, and reduce industrial waste.

## 2. Materials and Methods

### 2.1. Materials and Chemicals

The defatted chia flour is supplied by The Chia Company (Melbourne, Australia). All the chemicals used were of analytical grade from Sigma Aldrich (St. Louis, MO, USA) and Merck (Darmstadt, Germany).

### 2.2. Preparation of Chia Protein Isolate

Before protein isolation, the DCF sample was first de-mucilaged using the method outlined by Khushairay et al. [19]. Three different batches of DCF were mixed prior to the process. The mixed DCF was dissolved in distilled water (1:20 *w*/*v*), agitated for 30 min, and centrifuged at 10,000× *g* for 15 min to remove the mucilage. The mucilaginous intermediate phase of the slurry was removed, while the chia protein was recovered in the upper aqueous and bottom phases. The chia protein isolate (CPI) was obtained using alkali extraction-isoelectric precipitation (AE-IEP) according to the method outlined by Lan et al. [20] with slight modifications. The de-mucilaged DCF was disseminated into pH alkaline water (pH 8.5, 10.0, and 12.0) based on a DCF-to-water ratio of 1:20 for protein extraction [21]. The slurry was agitated using a magnetic stirrer at ambient temperature for one hour to solubilize the protein, followed by centrifugation (Allegra 64R Centrifuge, Beckman Coulter Inc., Indianapolis, IN, USA) for 30 min at 10,000× *g*. The supernatant was collected, filtered with filter paper (Whatman No. 4), and further acidified to pH 3.5 to precipitate the protein [21]. The centrifugation method was used to recover the protein precipitate, which was then washed thrice with 50 mL of Milli-Q water before being resuspended in Milli-Q water at pH 7.0. The mixture was stirred for 10 min to solubilize the protein. The protein was lyophilized using a freeze dryer (Scanvac CoolSafe Touch 110-4, Labogene, Allerod, Denmark.). Before further analysis, the lyophilized CPIs were stored in airtight packaging at 4 °C until further analysis.

### 2.3. Nutritional Properties of Chia Protein Isolate

#### 2.3.1. Proximate Composition, Extraction, and Protein Recovery Yield

The proximate composition was determined according to the Association of Official Analytical Chemists International (AOAC) [22]. The analyses were done accordingly: protein (method number 920.87, N × 6.25), lipids (method number 920.85), moisture (method number 935.29), ash (method number 932.03), and crude fibre (method number 962.09). The nitrogen-free extract (NFE) content was obtained by subtracting moisture, lipid, crude protein, crude fibre, and ash from 100%. The CPI (dry matter) extraction yield and protein recovery yield (%) were calculated using Equations (1) and (2), respectively.
(1)$$Extraction\;yield\left(\%\right)=\frac{Weight\;of\;lyophilized\;CPI\;powder\;(g)}{Weight\;of\;DCF\;powder\;(g)}\times 100$$
(2)$$Protein\;recovery\;yield\;(\%)=\frac{Protein\;content\;in\;lyophilized\;CPI\;powder\;(g)}{Protein\;content\;in\;DCF\;powder\;(g)}\times 100$$

#### 2.3.2. Protein Fractions

Proteins were extracted sequentially based on solubility differences according to Osborne classification using a modification method reported by Vazquez-Ovando et al. [23]. A 100 g demucilage DCF was suspended in 500 mL of distilled water before being extracted on a magnetic stirrer (Cimarec™ Digital Stirring Hotplates, Thermo Scientific, Waltham, MA USA) for 2 h at 4 °C. The suspension was centrifuged (Allegra 64R Centrifuge, Beckman Coulter Inc., Indianapolis, IN, USA) for 30 min at 950.3× *g*. The supernatant was decanted (designated as albumin fraction), and the residue was further extracted with 10 mL of 100 g/kg NaCl for 2 h and centrifuged to yield the globulin fraction. The resulting residue was then further extracted with 10 mL of a 700 g/L aqueous isopropanol solution under constant stirring. The supernatant was collected and designated as prolamin. The residue was resuspended in 10 mL of 0.1 M NaOH solution, centrifuged, and the separated supernatant was collected as glutelin. The residue after extraction was oven-dried at 90 °C for 6 h. All the protein fractions were lyophilized (Scanvac CoolSafe Touch 110-4, Labogene, Allerod, Denmark) and stored at 4 °C for further analysis.

#### 2.3.3. Amino Acid Profile

The amino acid profile of samples was measured according to the method reported by Abdul Manan et al. [24] with slight modification. The CPI samples were kept at 6 M hydrochloric acid (containing 0.1% phenol) at 110 °C for 24 h for protein hydrolysis. Cys and Met were estimated after pre-hydrolysis with performic acid oxidation before HPLC analysis, whereas Trp was determined by alkaline hydrolysis. The amino acids were derivatized in a heating block at 55 °C for 10 min and were separated using a C18 AccQ-Tag amino acid analysis column (3.9 × 150 mm, Waters, Milford, MA, USA) with a flow rate of 1 mLmin^−1^ and temperature-control of 37 °C. The UV detector’s wavelength for peak identification was 280 nm, while the fluorescence detector’s excitation and emission wavelengths for amino acid quantification were 250 and 395 nm, respectively. The amino acid compositions were computed based on the peak area relative to the standard. Amino acids were expressed in % of amino acids in the protein sample.

### 2.4. Functional Properties

#### 2.4.1. Protein Solubility

The soluble protein component of samples was determined according to the method of Nishanti et al. [25] with minor modifications. In brief, 12 mL of protein solutions (0.25% ww^−1^) was centrifuged (Allegra 64R Centrifuge, Beckman Coulter Inc., Indianapolis, IN, USA) at 4180× *g* for 10 min at room temperature. The protein concentration of the supernatant and the whole solutions were determined using the Biuret method. The protein solubility was calculated using Equation (3) as follows:(3)$$Protein\;solubility\left(\%\right)=\frac{Protein\;concentration\;of\;supernatant}{Total\;protein\;concentration\;of\;the\;whole\;solution}\times 100$$

#### 2.4.2. Water and Oil Absorption Capacities

The water and oil absorption capacities (WAC and OAC, respectively) were determined according to the method of Piornos et al. [26]. An 0.5 g sample was mixed with 5 mL distilled water (WAC) or 5 mL refined soybean oil (OAC) for 30 min. The slurry was centrifuged at 8667× *g* for 10 min (Allegra 64R Centrifuge, Beckman Coulter Inc., Indianapolis, IN, USA), and the unabsorbed water/oil was decanted and weighed. The supernatant was recovered, and its mass was measured. The water and oil absorption values were calculated using Equation (4) and were expressed as mass (g) of water or oil per unit mass (g) of protein.
(4)$$Water/Oil\;absorption\;capacities=\frac{W_{0}-Ws}{W_{0}}$$
where the *W*_0_ is the mass values of water/oil used (g) and *Ws* is the recovered water in the supernatant/unabsorbed oil (g).

#### 2.4.3. Emulsifying Properties

The emulsifying capacity (EC) and emulsion stability (ES) were determined by Drozlowska et al. [27] with some modifications. Aliquots (5 mL) of 10 mgmL^−1^ protein solutions at their corresponding pH value (pH 2–11) were thoroughly mixed with 5 mL of soybean oil using an Ultra-Turrax homogenizer (T25, IKA WERKE, Staufen, Germany) at 950.3× *g* for 1 min. The height of the emulsified layer and the total content of the tube were measured after 24 h. The emulsifying activity and emulsion stability were calculated using Equation (5) as follows:(5)$$Emulsifying\;capacity=\frac{H2}{H1}\times 100$$
where the *H*2 is the height of emulsified layer in the tube (mm) and *H*1 is the height of the total contents in the tube (mm). The emulsion stability was determined by measuring the total height after 5 to 120 min of the emulsion formation.

#### 2.4.4. Foaming Properties

The foaming capacity (FC) and stability (FS) were studied according to the method of Sha et al. [28] with slight modifications. Samples were dissolved in distilled water (1% of ww^−1^] protein equivalent, pH 2–11) using an Ultra-Turrax homogenizer (T25, IKA WERKE, Staufen, Germany) at 950.3× *g* for 1 min. Volumes were recorded before and after homogenization, and the foam capacity/stability was calculated using Equation (6):(6)$$Foam\;capacity/stability=\frac{V2-V1}{V1}\times 100$$
where *V*2 is the volume of suspension plus the foam (mL) and *V*1 is the initial volume of protein suspension (mL). The foam stability was determined by measuring the total volume after 5 to 120 min of the foam formation.

#### 2.4.5. Antioxidant Properties

DPPH radical scavenging assay

The DPPH scavenging activity of samples was measured using the method described by Shao et al. [29] with a slight modification. Briefly, a 70 μL of each sample was mixed with 70 μL of DPPH solution (150 μM in MeOH) and standing at room temperature for 30 min in the dark before measuring the absorbance using a microplate reader (Tecan GENios, Austria GmbH, Grodig, Austria) at 517 nm. The DPPH radical scavenging activity was calculated using Equation (7), and the result was expressed in % of the DPPH scavenging activities.
(7)$$DPPH\;Radical\;scavenging\;activity=\frac{Absorbance\;control-Absorbance\;sample}{Absorbance\;control}\times 100$$

ABTS radical scavenging assay

The ABTS radical stock solution was prepared according to the method described by Zielinska et al. [30] with slight modification. The ABTS solution and K_2_S_2_O_4_ were reacted in the dark for 12 h at 25 °C before diluting with phosphate buffer (0.7 ± 0.02 at absorbance 734 nm). A 50 μL sample and 150 μL of ABTS^+^ were mixed and stood for 5 min, and the absorbance was determined by a microplate reader (Tecan GENios, Austria GmbH, Grodig, Austria) at 734 nm. The ABTS radical scavenging activity was calculated using Equation (8), and the result was expressed in % of the ABTS radical scavenging activities.
(8)$$ABTS\;radical\;scavenging\;activity=\frac{Absorbance\;control-Absorbance\;sample}{Absorbance\;control}$$

Ferric reducing antioxidant power (FRAP) assay

The procedure described by Wootton-Beard et al. [31] was followed to determine the FRAP activity of the samples. Briefly, the FRAP reagent contained: 2.5 mL of 10 mmolL^−1^ TPTZ solution in 40 mmolL^−1^ HCl, 2.5 mL of 20 mmolL^−1^ FeCl_3_, and 25 mL of 0.3 molL^−1^ acetate buffer (pH 3.6). An aliquot of 40 μL of the sample was mixed with 0.2 mL distilled water and 1.8 mL FRAP reagent. The absorbance of the reaction mixture was spectrophotometrically measured at 593 nm after incubation at 37 °C for 10 min using a microplate reader (Tecan GENios, Austria GmbH, Grodig, Austria). Results were expressed in absorbance value.

#### 2.4.6. Antihypertension Properties

The ACE inhibitory (ACE-i) activity was determined using the method of Luo et al. [32]. Briefly, a mixture of 50 μL of sample solution and 50 μL of ACE solution (25 mUmL^1^) was pre-incubated at 37 °C for 10 min, followed by incubation with 150 μL of the substrate (8.3 mM HHL in 50 mM sodium borate buffer containing 0.5 M NaCl at pH 8.3) at 37 °C for 30 min. Distilled water was used as the blank and control, while Captopril (1 mgmL^−1^) was used as positive control. The reaction was stopped by the addition of 250 μL 1 M HCl, and the resulting hippuric acid was extracted by the addition of 500 μL of ethyl acetate. After centrifugation (3000 rpm, 10 min) (Allegra 64R Centrifuge, Beckman Coulter Inc., Indianapolis, IN, USA), a 200 μL of the upper layer was transferred into a glass tube and dried for 15 min at 90 °C. The hippuric acid was re-dissolved in 1 mL of distilled water, and the absorbance was measured at 228 nm using a fluorescence microplate reader (Tecan Austria GmbH, Grodig/Salzburg, Austria). The ACE-I activity was calculated using Equation (9), and the result was expressed in % of ACE-I inhibition.
(9)$$ACE-I\;activity=\frac{Absorbance\;sample-Absorbance\;control}{Absorbance\;sample-Absorbance\;blank}\times 100$$

### 2.5. Statistical Analysis

All experiments were carried out in triplicates. The Shapiro–Wilk test was employed to determine the normality of the experimental data. The data obtained from all the analysis were normally distributed and were subjected to a one-way analysis of variance (ANOVA). The differences among means were assessed using the Duncan multiple range test. The data were analyzed using SAS version 9.4 (Statistical Analysis Software) and were expressed as the mean ± standard deviation with *p* < 0.05, determining the significant differences between data.

## 3. Results and Discussion

### 3.1. Proximate Compositions, Protein Yield, and Fractions

The total extraction, protein recovery, and proximate composition of DCF and CPIs extracted at different alkaline pH were presented in Table 1. In accordance with the finding of earlier research, the extraction coefficient is highly dependent on the extraction pH, where elevating the pH will increase extraction yield [15,33]. Increasing the alkaline extraction pH from 8.5 to 12.0 increased the total extraction yield from 15.29 to 19.01%. Similarly, the protein recovery yield increased significantly from 44.29% at pH 8.5 to 59.63% at pH 12. 

According to Table 1, the total protein content of DCF was 37.89%, comparable to those 34.01 and 36.5% reported by Segura-Compos [34] and Coelho and Salas-Mellado [35], respectively. The protein concentration increased significantly after protein isolation proportionated to the increasing extraction pH for CPI-8.5, CPI-10.0, and CPI-12.0 with values of 69.75, 73.10, and 74.53%, respectively. The lipid content of protein isolates can be regarded as interference induced by the addition of sodium hydroxide, which can produce saponification reactions [35]. However, the protein isolation method via alkaline extraction in this study successfully reduced the lipid content to the range of 0.69 to 0.79%. The moisture content of CPIs varied between 4.39 to 4.54%. The CPIs extracted at high alkaline pH (pH 10 and 12) had a higher ash concentration, likely due to salt formation during protein precipitation at the isoelectric point. The crude fiber content of DCF was high (20.17%) due to the natural presence of chia mucilage. The protein isolation eliminated the mucilage component and reduced the fiber concentration to 0.29–0.31%. 

The fractionation of protein based on solubility discovered that the most dominant protein fraction in DCF and CPIs was glutelin (44.79% to 79.95%), consistent with the previous studies by Segura-Compos [34] that reported that the most abundant fraction in chia protein isolates was glutelin (42.94%). However, Sandoval-Oliveros and Paredez-Lopez [9] reported a contrary finding, mentioning that the glutelin proportion found in their chia protein isolates sample was low (14.5%). Nevertheless, it is obvious that the chia seed protein might differ depending on the botanical sources, the crop and cultivation environment, handling, and preparation techniques, and many more factors. 

### 3.2. Amino Acids Profiles

The amino acid composition has a potential role in the bioactivities of protein isolates. Table 2 demonstrates the amino acid profiles of DCF and CPIs. In general, there was an abundance concentration of Glu, Arg, Asp, Ser, and Phe in DCF and all the CPIs samples. A similar finding has been reported by Coelho and Salas-Mellado [35], Villanueva-Lazo et al. [36], Nitrayoya et al. [37], and Sandoval-Oliveros and Peredez-Lopez [9]. The DCF sample has a total of nine essential amino acids (EAA), the most prominent of which was Phe (5.94%). His, Val, Ile, Leu, Phe, Met, and Trp exhibited an increase in extraction efficiency throughout the isolation procedure, whereby Thr and Lys concentrations were reduced. According to Zhu et al. [38], the concentration of Thr, Ser, and Lys were reduced after the protein isolation process due to the decomposition of the compounds as a result of alkali treatment. The concentration of all the EAA in DCF and CPIs samples were adequate/more remarkable than the daily requirement pattern stated by WHO/FAO/UNU [39], except for Ile (in DCF, CPI-8.5, and CPI 10.0), which was expressed in a slightly lower concentration. However, the concentration of Ile in CPI-12.0 (1.97%) is comparable to needs (2.0%). It was found that the total proportion of EAA in DCF and CPIs was about twice as much as required, suggesting that the DCF and CPIs were an excellent source of EAA for daily consumption. These results were comparable to those obtained by chia protein isolates (37.7%), sesame protein hydrolysates (39.48%), and spirulina protein hydrolysates (31.16%) [35,40,41], but slightly inferior to mung bean protein isolates (43.51%) and jackfruit protein isolates (48.43%) [42,43]. 

According to Table 2, the total concentration of EAA was significantly increased (*p* < 0.05) from 34.76% (DCF) to 34.88% (CPI-8.5), 35.69% (CPI-10.0), and 36.87% (CPI-12.0). These data indicated that the extraction pH significantly affected the amino acids compositions following a previous study by Olivos-Lugo et al. [13], which reported that the EAA increased from 41.8% (defatted chia flour) to 42.8% (chia protein isolate), but in contrast with Coelho and Salas-Mellado [35] who claimed that the concentration reduced from 38.3% (defatted chia flour) to 37.7% (chia protein isolates). The hydrophobic amino acid (AAH) and aromatic amino acids (AAR) contents were higher in CPI-12.0 (33.81 and 15.54%, respectively) when compared to CPI-10.0 (32.27 and 14.97%, respectively), CPI-8.5 (31.15 and 14.42%, respectively), and DCF (30.63 and 13.97%, respectively), which could be attributed to the interactions of inter- and intra-polypeptides. A peptide’s function and bioactivity are profoundly affected by the types and numbers of amino acids it contains. Previous research has revealed that numerous amino acids and their derivatives, including hydrophobic amino acids Leu, Tyr, Val, Ala, and Pro and aromatic amino acid His, have antioxidant properties [44]. Moreover, it has been reported that hydrophobic amino acids located at the C-terminus of antihypertensive peptides with the presence of Val, Ala, Leu, Tyr, and Phe in the sequences are significant for the antihypertensive activity of the peptides [44,45]. Therefore, the high concentration of AAH and AAR in CPIs may be a favorable determinant of a great source of antioxidant and antihypertensive agents. 

### 3.3. Protein Solubility 

Solubility is a prerequisite techno-functional quality of proteins and a key determinant of its applications since many of their functional properties are dependent on the capacity to hydrate and solubilize in water/buffer [15]. The protein solubility of CPIs is depicted in Figure 1A. The arrangement of protein solubility in descending order are as follows: CPI-8.5 > CPI-10.0 > CPI-12.0 > DCF. Besides the influences of the hydrophilic and hydrophobic balance of the protein molecule, protein solubility is also affected by the composition of polar or non-polar amino acids mainly on the molecular surface, which controls the thermodynamics of protein–protein and protein–solvent interactions [46]. 

It is well established that the denaturation of protein during alkaline and acid treatments leads to the formation of protein aggregates. Protein extracted at high alkaline pH may have resulted in a higher degree of protein denaturation, producing low molecular weight protein fractions, thus increasing the exposure of more hydrophobic groups and decreasing the surface polarity [47]. Therefore, the protein solubility of CPI-10.0 and CPI-12.0 resulted in being significantly lower compared to CPI-8.5. Lopez et al. [48], who reported a similar solubility profile on chia protein isolates, and Ruiz et al. [33] on quinoa protein isolates with the lowest protein solubility, were obtained at their highest alkaline extraction pH. Furthermore, the higher concentration of hydrophobic and aromatic amino acids and the reduced proportion of hydrophilic amino acids (Arg, Lys, Asp, and Glu) in CPI-10.0 and CPI-12.0 (Table 2) may also have contributed to their lower protein solubility potential. Generally, the hydrophilic amino acids are more water interface-oriented, whereas the hydrophobic amino acids are interred within the protein’s hydrophobic core. The exposure to hydrophobic amino acids at the protein surface resulted in a hydrophobic environment that limits protein solubility [6].

The solubility of DCF and CPIs was investigated in an aqueous solution over a pH range of 2 to 11 to imitate the pH environment of the actual food matrix. The protein solubility values of all CPIs samples were observed ranging from 6.2 to 72.31%, with an inverse bell-shaped curve. The lowest protein solubility values for all samples were observed to occur consistently at pH 4 and 5 (Figure 1A), which could be due to the protein aggregation and precipitation as it moved closer to the isoelectric point. To be soluble, proteins should be able to interact extensively as possible with the solvent/solution. At the isoelectric point, proteins are electrically neutral as the negative and positive charges are canceled, repulsive forces are reduced, and the attraction forces are predominated. At this point, the hydrophobic interaction between neighboring leads to aggregation, and molecules tend to bind and precipitate, resulting in insolubility [6,49,50]. The interaction strength between proteins gradually decreases as the net protein charge increases at points away from the isoelectric point. Proteins become more negatively charged as the pH rises due to the carboxyl group ionization and amine group deprotonation. The increases in electrostatic repulsion between negatively charged proteins in more alkaline conditions improve the protein–water interactions as well as protein solubility [6]. This result might also be related to the higher globulin content in CPI-10.0 and CPI-12.0 compared to CPI-8.5 (Table 1), as it has been reported to have low solubility at pH 4 to 6 [51]. Previous reports on cumin seed [52], quinoa [33], and *A. trifoliata* var. *australis* seed [53] showed a similar trend for protein solubility. 

A protein’s water absorption capacity (WAC) is a measurement of its ability to associate gravity with water. It is likely related to the products’ texture, mouthfeel, and viscosity [54]. In contrast, the oil absorption capacity (OAC) is explained by the ability to bind with oil. It is closely related to flavor retention, shelf life, and emulsifying properties [55]. The WAC and OAC of the DCF and CPIs were presented in Figure 1(B). The WAC of DCF in this study was relatively higher (23.90 gg^−1^) compared to the CPI from a previous study by Mohammed Osama et al. [10] (8.5 gg^−1^) and was comparable to Coelho and Salas-Mellado [35] (~22 gg^−1^). The high WAC of DCF could be attributed to the high presence of fiber (20.17%) (Table 1), as this component significantly increased the WAC. The alkaline protein extraction treatment reduced the WAC of CPIs by approximately 6 times, with the least being CPI-12 (5.66 gg^−1^). This outcome could be attributable to the low fiber content (Table 1) and polar amino acids Cys and Tyr (Table 2) in CPIs compared to the DCF. The WAC of CPIs found in this study, however, was higher than the previously reported soy protein isolate (4.39 gg^−1^), cowpea protein isolates (1.38 gg^−1^), mung bean protein isolate (1.55 gg^−1^), and quinoa protein isolates (1.7–1.9 gg^−1^) [55,56]. 

According to Figure 1B, the OAC of DCF was 8.23 gg^−1^, significantly higher than all the CPI samples. CPI-12.0 revealed the highest OAC value for CPIs (7.55 gg^−1^), followed by CPI-10.0 (6.52 gg^−1^) and CPI-8.5 (5.24 gg^−1^). The OAC is closely related to the lipid–protein interaction via the binding of the non-polar side chains of amino acids to the aliphatic lipid chains [6,13]. In contrast, the concentration of non-polar amino acids (Leu, Ile, Pro, Trp, Val, Phe, and Met) in this study was inversely related to the OAC values. The low OAC value for CPIs could be attributed to the protein denaturation by alkaline treatment during extraction, which exposed more hydrophobic sites, making them available to react with lipids. These findings substantiated the high proportion of hydrophobic amino acids in CPIs extracted at a more alkaline pH (Table 2). According to Deng et al. [54], factors such as the amino acid content, surface hydrophobicity, and the protein extraction method used can all influence the OAC of protein. The OAC values of CPIs revealed in this study were comparable to those previously reported by Lopez et al. [48], which ranged from 6.1 to 7.1 gg^−1^ and were slightly higher than the CPI described by [57], with a value of 4.04 gg^−1^. Compared to other plant protein sources, the CPIs in this study were also higher than quinoa isolates [57], soy protein isolates [55], flaxseed protein concentrated [58], sour cherry kernels protein concentrated [59], and chickpea protein concentrated [60].

### 3.4. Emulsifying Properties

Emulsion capacity (EC) is the ability of an emulsifying agent to generate a water-in-oil dispersion, whereas emulsion stability (ES) is the rate at which it degrades over time. The EC and ES of DCF and CPIs at different pH over a period of 120 min were demonstrated in Figure 2. According to Figure 2A, DCFs offered better EC than CPIs, indicating that the ability of the CPIs to reduce the interfacial tension and create a protective barrier around the oil droplet was somewhat limited. CPIs demonstrated poorer emulsifying capacity (CPI-8.5 > CPI-10.0 > CPI 12.0), most likely because the alkaline treatment during the extraction process resulted in significant protein denaturation and insolubility, thus reducing their emulsifying capabilities. All samples illustrated U-shaped curves within the pH range of 2 to 11 (Figure 2A), indicating a connection between pH, emulsifying ability, and protein solubility. Since the emulsifying capacity of soluble proteins is dependent on the hydrophilic-lipophilic balance, which is influenced by pH, it has been established that pH significantly affects emulsifying activity [34]. Moreover, the emulsifying capacity relies on forming stable interfacial protein membranes. Therefore, the emulsion formation should generally increase with soluble protein concentration [50], as it enables enhanced interaction between the oil and the aqueous phase.

As demonstrated in Figure 2A, the DCF and CPIs have the lowest EC at pH 4 and 5 and start to improve at pH 7 to 11. This pattern was consistent with earlier research on protein isolates from Chinese quince seed [54], Camelina and Sophia [55], mung bean [61], and chickpea [62]. The lowest EC values of all samples were found at pH 4, which is thought to be closely connected to the weak electrostatic repulsion among the protein molecules, which induced aggregation and destabilized the interfacial membrane near the isoelectric point. The EC enhanced with the increase of hydroxide ions and reached the highest values at pH 11, which might be caused by the increase of molecular flexibility and surface hydrophobicity at higher pH conditions. 

The ES of all samples at all pH levels was monitored for 120 min to determine the ability of the emulsion to resist physicochemical changes over time. Results are demonstrated in Figure 2B–E. Among all samples, the DCF had the highest ES at pH 5 (82.05%), which was sustained for 10 min before decreasing. CPI-12.0, on the other hand, demonstrated the lowest ES at pH 2 (6.99% at 5 min) and continued to reduce over time (3.52% at 120 min). The ES of CPIs was low at the low pH levels and increased as the pH increased. The low ES at low pH is due to the presence of chloride ions, which reduced the protein’s net charge and increased interaction between emulsified droplets. By increasing the pH, the interface energy in the emulsion droplets was decreased due to the increase of coulombic repulsion among adjacent droplets, along with the charged protein molecules’ hydration, thus improving the ES [34,63]. 

### 3.5. Foaming Properties

Foaming capacity (FC) is the percentage increase in volume after suspension mixing and represents the interaction between a liquid solution and the air, whereas foam stability (FS) is defined as the length of time the foam maintains its volume after being formed [34,48]. There is a significant correlation between protein solubility and foaming capacity/stability as it coincides with the capability to decrease surface tension at the water–air interface. The increased protein solubility enhanced the interaction between water and proteins, which aids in unfolding the protein structure, and, thus, increases air encapsulation [64]. Figure 3 depicts the FC and FS of DCF and CPIs at different pHs over 120 min. Both FC and FS values of all samples were lowest at the least protein-soluble point (pH 4). This could be due to the fact that at this point, the ability of proteins to diffuse and form a bubble at the water–air interface was restricted. This behavior is attributed to the lack of protein solubility, which reduces the water–protein interactions and hinders the unfolding of protein structure, therefore reducing the air encapsulation [55,64]. The FC increases at pH below 4 and above 5, with the highest at pH 11, corresponding with the reported protein solubility (Figure 1A). The high FC at higher pH could be explained by the increases in protein net charge, which reduced the hydrophobic interactions, and enhanced the flexibility of protein, thus allowing the protein to efficiently permeate to the water–air interface, encapsulating air bubbles and accelerating the production of foam [65,66]. 

The FS of all samples at all pH levels was monitored over 120 min to determine the range of feasible bubble persistence. Generally, a protein with an excellent surface-active characteristic and sufficient intermolecular interaction has good foaming ability [59]. A significantly elevated FS at high alkaline pH could be ascribed to the improvement of the solubility and surface activity of the proteins. According to Figure 3, the CPI-12.0 show the highest FC (83.16%) and FS (83.10% measured at 5 min) both at pH 11, while the DCF has the lowest of both at pH 4 (FC: 3.24%; FS: 0.42% measured at 120 min). The FS of DCF and CPIs were gradually degraded over time, exposing the inability of the proteins to generate cohesive interfacial water–air films with adequate mechanical qualities, therefore limiting the foam’s stability [65]. 

### 3.6. Antioxidant and Antihypertensive Properties

The antioxidant activities of the chia protein isolates were determined via DPPH and ABTS radical scavenging assays and FRAP reducing power assays are depicted in Figure 4. The DPPH activities of CPIs varied from 23.23% to 42.48%, whereas the ABTS activity ranged from 31.66% to 66.25%. The radical scavenging activities increased with isolation pH up to pH 10. The highest radical scavenging potential was found in sample CI-10, approximately 5 times greater than DCF (as demonstrated by the DPPH assay) and 5.5 times higher (as revealed by the ABTS assay). However, the radical scavenging activities decreased significantly at pH 12, which the degradation effect might cause at high extraction pH. The correlation between the amino acid composition of the protein and antioxidant potential has been emphasized in the literature. It has been found that peptides with strong antioxidant activity contain a more significant proportion of hydrophobic amino acids than hydrophilic amino acids, which is regarded as the determining factor in peptide’s ability to scavenge radicals [67]. 

On the other hand, aromatic amino acids such as Tyr, His, Trp, and Phe are also proven to have a positive correlation with the antioxidant potential [68,69]. This clearly explains the higher radical scavenging activity (as determined by DPPH and ABTS) on CPI-10.0 compared to CPI-8.5. Even though the hydrophobic and aromatic amino acid concentration in CPI-12.0 was higher than in CPI-10.0 (Table 2), the scavenging capacity is shown to be lower. This may be due to the excessive protein denaturation at high pH levels producing a large amount of inactive free amino acids. According to Mori et al. [70], an extreme extraction condition such as high alkaline pH may affect the thermal, conformational, and functional characteristics of protein fractions, diminishing their nutritional value and destroying their bioactive components. The resulting isolates showed greater ABTS radical scavenging activities than DPPH radical scavenging activities. The mechanism of these two assays might be explained by the fact that ABTS radical scavenging is more applicable in the hydrophilic compound, whereas DPPH radical scavenging is significantly influenced by the hydrophobic system [71]. The FRAP activity was indicated via the absorbance value, which showed a pattern comparable to ABTS and DPPH activities. 

The antihypertensive activity of CPIs ranged from 25.68% to 35.67%, which was lower than 1 mgmL^−1^ captopril (99.02%), as illustrated in Figure 4D. The CPI obtained with pH 10 (CPI-10) exhibited the highest ACE-inhibitory potential (2.5 times higher than DCF), whereas the lowest value was obtained at pH 8.5 (25.68%). The amino acid composition and its sequences have been shown to significantly influence the ACE inhibitory action [72]. The ACE-inhibitory peptides typically contain hydrophobic amino acids at the N-terminus of their sequence [73,74]. The result could be explained by the high proportion of AHH in CPI-10.0 (Table 2). Again, even though CPI-12.0 has exhibited higher AAH concentration than CPI-10.0, the disruption of most bioactive amino acid chains by high alkaline treatment (pH 12) may be the limiting factor to the antihypertensive capacity. ACE inhibitors have been proven to improve oxidative stress by inhibiting the production of angiotensin II. The strong positive correlation between these dual bioactivities has suggested that the antioxidant capacities increase with the antihypertensive activities [68,75]. Thus, this finding suggested that the CPI-10.0 in this study could be a promising antioxidant and antihypertensive source that may be further exploited in the production of bioactive peptide-based functional food formulated for health purposes and preventing diseases.

## 4. Conclusions

The protein isolation of DCF via alkaline extraction at different pH significantly affects the nutritional and functional properties of the CPIs. Extraction of DCF at pH 12.0 have successfully achieved the highest extraction yield (19.10%), protein recovery yield (59.63%), and total protein content (74.53%), as well as improved the proportion of essential (36.87%), hydrophobic (33.81%), and aromatic amino acids (15.54%). The alkaline extraction process does not aid in the water-absorption capacity and oil-absorption capacity of the CPIs. Still, it improves the protein solubility, foaming capacity, and foam stability in various pH environments. Even though alkaline extraction does not enhance the emulsifying capacity and emulsion stability, the protein extraction at pH 10.0 successfully produced CPIs with the highest antioxidant (DPPH: 42.48%; ABTS: 66.23%; FRAP: 0.19) and antihypertensive (35.67%) capacities. The CPIs developed from the underutilized DCF have a high potential to be employed as a new natural antioxidant and antihypertensive ingredients for functional food and nutraceutical application. In addition, the finding of this study may maximize the exploitation of the DCF as well as lessen industrial waste. With enhanced nutritional and functional values, more research investigations, including in vivo experiments, are clearly needed to establish this health-promoting impact of protein isolates from chia on humans.

## Figures and Tables

**Figure 1 foods-12-03046-f001:**
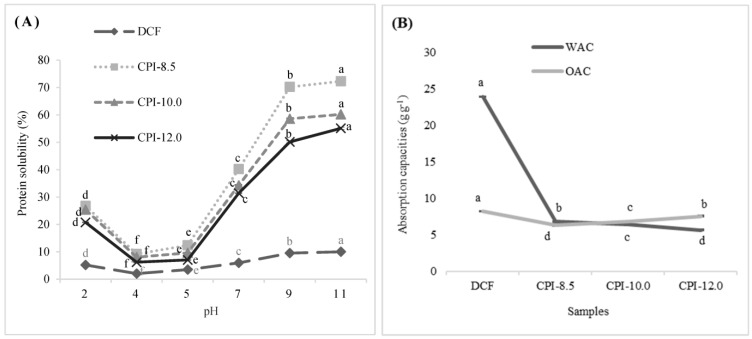
The protein solubility and absorption capacities of DCF and CPIs. (**A**): The protein solubility of DCF and CPIs at various pH levels; (**B**): The water absorption capacities (WAC) and oil absorption capacities (OAC) of DCF and CPIs. ^a–f^ Different superscript letters indicated statistical differences (*p* < 0.05). Data are the mean ± SD of three replications. DCF: Defatted chia flour; CPI-8.5: isolates using pH8.5; CPI-10.0: isolates using pH10.0; CPI-12.0: isolates using pH12.0.

**Figure 2 foods-12-03046-f002:**
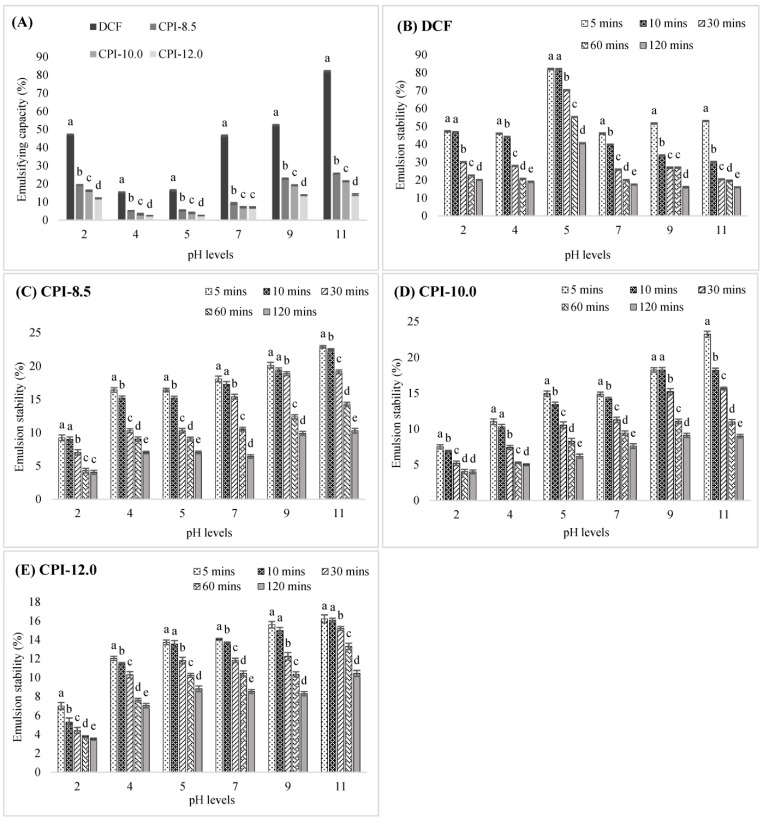
The emulsion properties of DCF and CPIs at different pH levels over a period of time. (**A**): The emulsifying capacity of DCF and CPIs; (**B**): The emulsion stability of DCF; (**C**): The emulsion stability of CPI-8.5; (**D**): The emulsion stability of CPI-10.0; (**E**) The emulsion stability of CPI-12.0. ^a–e^ Different superscript letters in the same pH levels are significantly different (*p* < 0.05). Data are the mean ± SD of three replications. DCF: Defatted chia flour; CPI-8.5: isolates using pH8.5; CPI-10.0: isolates using pH10.0; CPI-12.0: isolates using pH12.0.

**Figure 3 foods-12-03046-f003:**
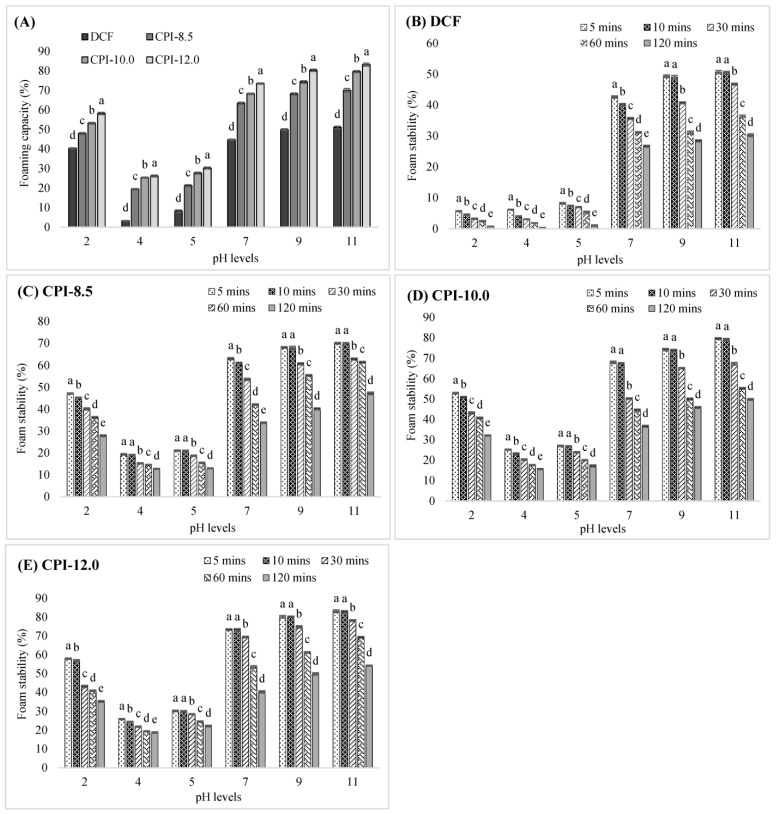
The foaming properties of DCF and CPIs at different pH levels over a period of time. (**A**): The foaming capacity of DCF and CPIs; (**B**): The foam stability of DCF; (**C**): The foam stability of CPI-8.5; (**D**): The foam stability of CPI-10.0; (**E**) The foam stability of CPI-12.0. ^a–e^ Different superscript letters in the same pH levels are significantly different (*p* < 0.05). Data are the mean ± SD of three replications. DCF: Defatted chia flour; CPI-8.5: isolates using pH8.5; CPI-10.0: isolates using pH10.0; CPI-12.0: isolates using pH12.0.

**Figure 4 foods-12-03046-f004:**
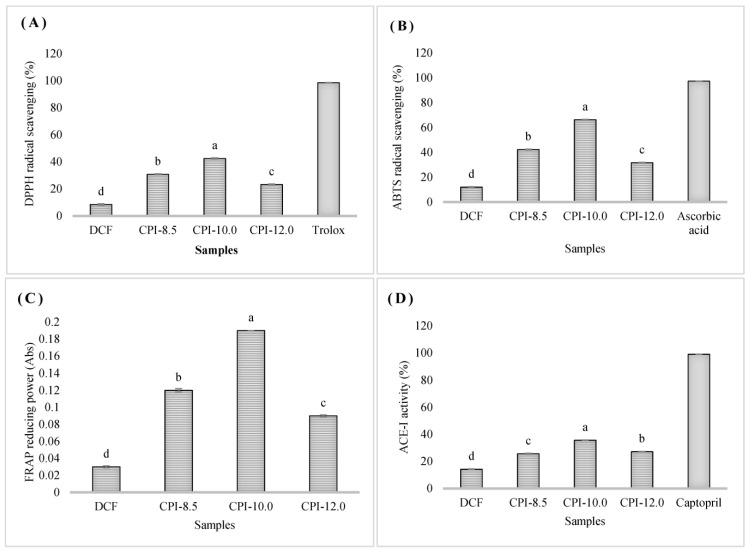
The antioxidant and antihypertensive activities of DCF and CPIs: (**A**) DPPH radical scavenging activities; (**B**) ABTS radical scavenging activities; (**C**) FRAP reducing power; (**D**) ACE-i capacity. ^a–d^ Different superscript letters are significantly different (*p* < 0.05). Data are the mean ± SD of three replications. DCF: Defatted chia flour; CPI-8.5: isolates using pH8.5; CPI-10.0: isolates using pH10.0; CPI-12.0: isolates using pH12.0.

**Table 1 foods-12-03046-t001:** The extraction yield, protein recovery yield, and proximate composition of DCF and CPIs.

		DCF	CPI-8.5	CPI-10.0	CPI-12.0
Extraction yield and protein recovery (%)					
Extraction yield			15.29 ± 0.09 ^c^	16.87 ± 0.16 ^b^	19.10 ± 0.11 ^a^
Protein recovery yield			44.29 ± 0.17 ^c^	52.31 ± 0.20 ^b^	59.63 ± 0.15 ^a^
Proximate composition (%)					
Crude protein		37.89 ± 0.25 ^d^	69.75 ± 0.23 ^c^	73.10 ± 0.17 ^b^	74.53 ± 0.21 ^a^
Protein fractions	Glutelin	44.79 ± 0.19 ^d^	79.16 ± 0.22 ^b^	78.89 ± 0.26 ^c^	79.95 ± 0.22 ^a^
	Globulin	26.32 ± 0.11 ^a^	5.65 ± 0.15 ^c^	6.04 ± 0.09 ^b^	6.15 ± 0.12 ^b^
	Albumin	25.22 ± 0.09 ^a^	6.89 ± 0.31 ^b^	6.58 ± 0.12 ^b^	6.72 ± 0.20 ^b^
	Prolamin	3.67 ± 0.13 ^c^	7.80 ± 0.17 ^b^	8.49 ± 0.12 ^a^	7.68 ± 0.21 ^b^
Lipid		15.12 ± 0.23 ^a^	0.79 ± 0.17 ^b^	0.71 ± 0.15 ^b^	0.69 ± 0.19 ^b^
Moisture		7.43 ± 0.11 ^a^	4.39 ± 0.10 ^b^	4.42 ± 0.11 ^b^	4.54 ± 0.15 ^b^
Ash		5.21 ± 0.14 ^c^	6.22 ± 0.09 ^b^	6.41 ± 0.10 ^a^	6.44 ± 0.12 ^a^
Crude Fibre		20.17 ± 0.12 ^a^	0.30 ± 0.02 ^b^	0.31 ± 0.04 ^b^	0.29 ± 0.04 ^b^
Nitrogen-free extract (NFE)		14.18± 0.17 ^c^	18.55 ± 0.14 ^a^	15.05 ± 0.14 ^b^	13.51 ± 0.15 ^d^

The values shown are the mean ± SD of three replicates. Values in the same row without a common ^a–d^ superscript letter are significantly different (*p* < 0.05). DCF: Defatted chia flour; CPI-8.5: isolates using pH8.5; CPI-10.0: isolates using pH10.0; CPI-12.0: isolates using pH12.0.

**Table 2 foods-12-03046-t002:** The amino acid profiles (%) of chia protein isolates (CPI) extracted at different extraction pH.

Amino Acids	DCF	CPI-8.5	CPI-10.0	CPI-12.0	^1^ FAO/WHO/UNU (Adult)
Essential amino acids (EAA)					
Ile	1.69 ± 0.08 ^c^	1.72 ± 0.05 ^c^	1.88 ± 0.05 ^b^	1.97 ± 0.04 ^a^	2.0
Leu	4.01 ± 0.05 ^c^	3.99 ± 0.07 ^c^	4.15 + 0.02 ^b^	4.29 ± 0.03 ^a^	3.9
Lys	4.64 ± 0.05 ^a^	4.59 ± 0.09 ^a^	4.45 ± 0.12 ^b^	4.39 ± 0.08 ^b^	3.0
Met	4.48 ± 0.04 ^d^	4.57 ± 0.07 ^c^	4.87 ± 0.06 ^b^	5.29 ± 0.05 ^a^	1.5
Phe	5.94 ± 0.06 ^d^	6.08 ± 0.05 ^c^	6.33 ± 0.04 ^b^	6.63 ± 0.06 ^a^	2.5
Thr	5.43 ± 0.09 ^a^	4.90 ± 0.11 ^b^	4.55 ± 0.06 ^c^	4.50 ± 0.09 ^c^	1.5
Val	4.00 ± 0.10 ^d^	4.23 ± 0.09 ^c^	4.41 ± 0.05 ^b^	4.60 ± 0.07 ^a^	2.6
His	3.52 ± 0.05 ^c^	3.61 ± 0.03 ^b^	3.77 ± 0.04 ^a^	3.80 ± 0.04 ^a^	1.0
Trp	1.05 ± 0.04 ^d^	1.19 ± 0.03 ^c^	1.28 ± 0.05 ^b^	1.40 ± 0.01 ^a^	0.4
Non-essential amino acids (NEAA)					
Ala	3.33 ± 0.06 ^a^	3.21 ± 0.04 ^b^	3.25 ± 0.10 ^b^	3.24 ± 0.05 ^b^	
Arg	12.69 ± 0.07 ^b^	12.93 ± 0.04 ^a^	12.97 ± 0.04 ^a^	12.02 ± 0.03 ^c^	
Asp	9.72 ± 0.08 ^a^	9.55 ± 0.06 ^b^	9.40 ± 0.06 ^c^	9.12 ± 0.05 ^d^	
Cys	0.88 ± 0.01 ^b^	0.93 ± 0.02 ^a^	0.90 ± 0.01 ^a^	0.94 ± 0.04 ^a^	
Glu	18.46 ± 0.07 ^a^	18.22 ± 0.06 ^b^	18.21 ± 0.10 ^b^	18.07 ± 0.09 ^c^	
Gly	5.58 ± 0.05 ^a^	5.51 ± 0.03 ^b^	5.19 ± 0.06 ^c^	5.09 ± 0.04 ^d^	
Ser	7.40 ± 0.03 ^a^	7.42 ± 0.05 ^a^	7.01 ± 0.05 ^b^	6.86 ± 0.04 ^c^	
Tyr	3.46 ± 0.07 ^c^	3.54 ± 0.05 ^b^	3.59 ± 0.03 ^b^	3.70 ± 0.03 ^a^	
Pro	3.72 ± 0.05 ^c^	3.81 ±0.04 ^b^	3.79 ± 0.05 ^b^	4.09 ± 0.07 ^a^	
Total EAA	34.76 ± 0.06 ^d^	34.88 ± 0.07 ^c^	35.69 ± 0.05 ^b^	36.87 ± 0.05 ^a^	18.4
Total ^2^ AAH	30.63 ± 0.06 ^d^	31.15 ± 0.06 ^c^	32.27 ± 0.05 ^b^	33.81 ± 0.05 ^a^	
Total ^3^ AAR	13.97 ± 0.06 ^d^	14.42 ± 0.04 ^c^	14.97 ± 0.04 ^b^	15.54 ± 0.03 ^a^	

The values shown are the mean ± SD of three replicates. Values in the same row without a common ^a–d^ superscript letter are significantly different (*p* < 0.05). ^1^ FAO/WHO/UNU: Energy and protein requirements (2007). ^2^ AAH: Hydrophobic amino acid (Ala, Val, Met, Ile, Leu, Phe, Pro, and Tyr); ^3^ AAR: Aromatic amino acid (Phe, His, Trp, and Tyr). DCF: Defatted chia flour; CPI-8.5: isolates using pH8.5; CPI-10.0: isolates using pH10.0; CPI-12.0: isolates using pH12.0.

## Data Availability

The data shown in this study are contained within the article.

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
