# Peer review of "The Nutritional and Functional Properties of Protein Isolates from Defatted Chia Flour Using Different Extraction pH"

_foods, 2023, doi:10.3390/foods12163046_

Round 1

Reviewer 1 Report

Revisions and comments

Dear Authors,

I commend your efforts in assessing the effect of different alkaline extraction protocols on the nutritional and functional properties of protein isolated from defatted chia flour. I consider that the methodology is clear and that the results are adequately presented and discussed.

I have some comments:

Abstract (lines 12-24):

The abstract does not describe the methods employed in the study. Please briefly add the main methods or treatments applied.

Materials and Methods Section, Preparation of Chia Protein Isolate:

In line 82, the authors state that DCF was dissolved in distilled water (1:20). Please modify (1:20) to (1:20 w/v) if it is the case.

Materials and Methods Section:

Please change rpm units to g units (line 118, page 3; line 163, page 4; line 175, page 4).

Materials and Methods Section, Antihypertension Properties:

In the results, the authors show that captopril was used as a positive control, but this is not declared in the material and methods section. Additionally, it is not described how captopril was prepared for the assay. Please state it.

Materials and Methods and Results Section, Statistical Analysis:

- Did you employ a normality test to analyze the data distribution of the study? If so, which one?

- The authors used a one-way analysis of variance followed by a Duncan multiple range test. Was this same analysis employed for all the data analyzed?

- The authors analyzed different data that came from other assays. Did all the data obtained from all the assays follow a normal distribution? Please declare.

- Why do the authors employ the Duncan multiple range test instead of the Tukey test? Please explain.

- As I can see in the different results presented by the authors, there are some comparisons that appear to be paired between the samples. For example, in Figure 2, the authors compare the emulsion stability between samples obtained at the same pH levels across time. If so, did the authors employ repeated measures one-way ANOVA instead of a regular one? Please declare.

Results section, Figure 1 (A):

In Figure 1(A), the authors compare the protein solubility of CPI obtained at pH 8.5, 10.0, and 12.0 in solutions at different pHs. Was the comparison of protein solubility performed between samples (CPI 8.5, 10, and 12) solubilized at different pHs (2, 4, 5, 7, 9, 11)? Did the authors compare the protein solubility of the same sample solubilized at different pHs? For example, was compared the protein solubility of CPI 8.5 at different pH (2, 4, 5, 7, 9, 11) compared?

Results section, Figure 2:

Figure 2, it is somewhat confusing the way that the results and the statistical comparisons are presented. For example, in Figure 2 Section B, the authors presented the emulsion stability of DCP samples obtained at different pH (2, 4, 5, 7, 9, 11) at different minutes (5, 10, 30, 60, and 120). In the X-axis, we can see the time (minutes), and in the Y-axis the emulsion stability (%). Was the main purpose of this analysis to compare the pair of DPC samples obtained at different pHs at different times? For example, was the aim to compare the DCP sample obtained at pH 2 across different times? (5, 10, 30, 60, 120). If that is the case, the figure could be easier to understand if the DCP samples obtained at different pHs were on the X-axis instead of at different times. In this way, the bars of each sample obtained at different pH levels across time could be grouped together instead of being separated.

Results section, Line 504:

Please correct Figure 1 (D) to Figure 4 (D).

Minor editing of English language required.

Author Response

Dear Reviewer, We highly value your comments to improve our manuscript.

Reviewer 2 Report

Here are the comments and suggestions: 

1. The title is clear and concise, effectively conveying the main focus of the study. However, it could benefit from including the term "protein isolates" to provide more specific information about the content of the paper.

2. The abstract provides a good overview of the study, including the objectives, methods, and key findings. However, it would be helpful to briefly mention the significance or implications of the results to give readers a better understanding of the study's potential impact.

3. The authors are advised to separate the Results and Discussion for better understanding.

4. The conclusion provides a summary of the main findings and their implications. However, it would be beneficial to include specific numerical values or ranges for the various properties discussed, as this would provide more concrete information for readers. 

5. Additionally, it would be helpful to briefly mention any limitations of the study or avenues for future research to provide a well-rounded conclusion.

6. Overall, the manuscript seems well-structured and the findings are clearly presented. However, to provide a comprehensive review, I would need to assess the full manuscript, including the methodology, results, and discussion sections.

Minor editing of English language required

Author Response

(The authors gave the same response as above.)

Reviewer 3 Report

The manuscript “The Nutritional and Functional Properties of Protein Isolated from Defatted Chia Flour Using Different Extraction pH” by Etty Syarmila Ibrahim Khushairay and collaborators describes a study aimed at investigating the effects of various pH values on the features of protein extracts from defatted chia flour, that is a waste from chia oil industry. For instance, the Authors analyze features such as the amino acid profiles, protein solubility, emulsifying, foaming antioxidant and antihypertensive properties.

This topic is of some interest because it falls under sustainable food alternatives. The study appears well designed and well performed. The presentation is clear and the conclusions are supported by the results.

These are some specific comments.

1. Introduction, lines 67-69. “Isoelectric precipitation is a typical method for isolating alkali-extracted plant proteins and preparing high-purity protein-rich product”

After the oil extraction from chia seed, can the resulting waste be used as it is? For instance by producing a flour from the chia waste? Why the chia proteins have to be extracted? Was all the waste edible?

 2. Materials and Methods, line 80. “…de-mucilage…”

Should it be "de-mucilaged"?

 3. Line 108. “…yeild…”

Should it be yield? Please check all text for typing mistakes.

4. Lines 131-132. “The amino acids were synthesized….”

What do the authors mean with "synthesized"? Should it be "derivativeized"?

5. Table 2. A measure unit should be added to the headers. Which is the meaning of numbers? Is it "%"? Which is the unit of the numbers in the column "FAO/WHO/UNU"?

6. Table 2 footnotes. “Values in the same row without a common.” Which is the meaning? Is a verb missing?

7. Lines 340-342. “The low protein solubility values at pH 4 and 5 could be explained by protein aggregating and precipitating near the isoelectric point”

Which chia proteins do have pI near pH 4 and 5? This information could be added in the text.

8. Lines 413-415. “This pattern was consistent with earlier research on Chinese quince seed protein isolates [54], Camelina and Sophia protein isolates [55], mung bean protein isolates [61], and chickpea protein isolates [62].”

This sentence can be modified in order to avoid the repetition of “protein isolates”.

I suggest something like: “This pattern was consistent with earlier research on protein isolates from Chinese quince seed [54], Camelina and Sophia ....and chickpea [62].

Anyway, what are Camelina and Sophia? Apple cultivars?

9. Line 542. Please check the verb “The data showed…”

Should it be “The data shown…”?

 In my opinion, English language is quite good. I suggest checking the text because there are minor mistakes

Author Response

(The authors gave the same response as above.)

Round 2

Reviewer 1 Report

The authors addressed all the comments properly.

Minor editing of the English language is required.